# The Effect of a High Amount of Micro-Fillers on the Long-Term Properties of Concrete

**DOI:** 10.3390/ma12203421

**Published:** 2019-10-18

**Authors:** Alena Sičáková, Matej Špak

**Affiliations:** Faculty of Civil Engineering, Technical University of Košice, Vysokoškolská 4, 042 00 Košice, Slovakia; matej.spak@tuke.sk

**Keywords:** concrete, micro-filler, brick powder, concrete powder, glass powder

## Abstract

Concretes in which a large portion of fine natural aggregate is replaced with inert mineral powders would offer both economic and ecological benefits for the concrete industry, and they represent eco-friendly materials. Moreover, using the powders having potential pozzolanic effect could have positive extra effect on the properties of concrete. This paper analyses the impact of a high dosage of three kinds of micro-fillers (brick, concrete and glass powders) on the properties of concrete over a three-year period. Microfillers were applied as 40% replacement of 0/4 aggregate by volume. Samples having high dosage of micro-fillers and thus a higher binder volume achieved excellent values of both compressive (from 31 to 48 MPa in 28 days, and from 67 to 93 MPa in three years) and flexural strength (from 6.3 to 8.4 MPa in 28 days, and from 7.1 to 11.1 MPa in three years). Both samples with brick powder and concrete powder achieved the biggest strength values; however, due to better performance in durability parameters (capillary water absorption coefficient and density), sample prepared with glass powder can be identified as having the biggest potential for intended use.

## 1. Introduction

Fine-grain mineral material-additives play an irreplaceable role in the current technology of cement-based building mixes such as concrete. Materials like fly ash, silica fume and ground granulated blast furnace slag are used already, while they are considered standard. Moreover, a number of other materials are being investigated worldwide, like metakaolin, rice husk ash, fine particles of various rocks, powdery glass, bricks, and others. These materials have certain properties that make them affect the character of concrete in different ways [1]. However, the technical conditions for their use in concrete have still not been determined with sufficient precision.

According to [1], mineral additives may be categorized as follows—chemically active mineral admixtures (highly reactive pozzolan) and micro-filler mineral additives (low to moderately reactive pozzolan). The character depends on physical (particle size, particle shape, specific surface area, etc.) and chemical properties (chemical and mineralogical/phase composition, ratio of hydraulic oxides); the source and process from which these components are obtained are responsible for the particular character.

### 1.1. Pozzolans

Pozzolans are presented as siliceous or aluminosilicate materials that have little or no cementitious ability, but in fine-grained form and in the presence of moisture, react with calcium hydroxide at normal temperatures to produce compounds having cementitious properties [2]. To have pozzolanic properties, standards for fly-ash [2,3] requires that summation of the SiO_2_, Al_2_O_3_, and Fe_2_O_3_ should be more than 70%. Some authors state that it is possible to replace cement with a pozzolan beyond 30% by weight of cement, as supplementary cementitious material (SCM) [4]. 

As for inert micro-fillers, their small particle size and high specific surface area are advantageous for the production of high-density and tight concrete, as these have some influence on the strength and durability of the concrete [1].

One reason for verifying the applicability of a wide range of different powdery materials is that they are usually a form of waste product. Due to the sustainability principles, which are becoming a priority all over the world, greater emphasis is placed on the potential of waste recycling for the production of green cement-based building materials [5,6]. In recent years, the use of alternative materials has been increasingly considered in order to reduce carbon dioxide emissions and landfill pollution [7]. Thanks to reactive properties of some additives, they can be applied as a partial replacement of cement; similarly, micro-fillers can be used as partial replacement of fine natural aggregates and can thus offer important environmental benefit.

The consequence of intensive urbanization is the creation of a huge amount of construction and demolition waste, while increasing the consumption of natural raw materials [8]. In the built-up areas, 30–40% of the waste from demolition was recorded out of the total waste [9]. Therefore, materials from the category of demolition waste are of interest in terms of their recyclability, while in fine-grained form they could have the potential for just cement-based construction mixtures, as seen in the case of fired clay brick, concrete, and glass. For research purposes, powdery materials are usually prepared by grinding.

### 1.2. Brick Powder (BP)

Brick powder is often investigated as a potential cement replacement to create blended cement, than as a micro-filler. The pozzolanic activity of BP strongly depends on grain size, while authors give different values to this feature [8,10,11,12,13]. Studies show that brick powder is a promising material for the production of pozzolanic cement [10], mortar [11], and concrete [12], provided the material is of a particle size less than 60 μm and the replacement ratio is up to 25%. Authors state that brick powder reduces the alkali–silica reactions of cement-based mortar and concrete; further, when used with the optimum mixture ratio, it increases the strength of concrete compared to the control concrete, although it increases the setting time. Sabir et al. [14] report a low initial compressive strength of concrete when partially replacing Portland cement with ground red bricks, but similar or greater later strength (after 90 days) than recorded for PC-based samples.

Rahhal et al. [13] tested local materials—powdered ceramic waste obtained by grinding the scrap discarded from local brick factories, Argentine d(0.9) = 64.7 μm and Czech d(0.9) = 41.0 μm—for the production of blended cements (8–40% substitution of local Portland cements, by mass). They characterize the time behavior of BP; it acts as a filler at an early age, then with hydration process the calcium hydroxide is partially consumed due to pozzolanic activity. As a result, hydrated calcium aluminates grow depending on the age and carbonates present. Finally, cements blended with fired brick powder had low compressive strength at early ages but comparable strength-class at a later age. Baronio and Binda [15] highlight the importance of chemical/phase composition for the pozzolanic activity of bricks. All bricks may not have pozzolanic properties. The firing temperature, as well as content of clay minerals depending on the geological origin of raw materials, is important.

Li et al. [8] tested BP of d(0.85) = 50.0 μm as micro-filler to replace the quartz sand (20, 40, 60, 80, and 100%) in decorative plasters—i.e., in terms of cement, it was an addition. Authors report that even the sample prepared with 100% BP satisfies the requirements of local standards. The compressive and flexural strengths of plasters made with BP were improved, the density was increased for the improvement of the plaster microstructure, and the plasters generally exhibited better performance.

### 1.3. Recycled Concrete Powder(RCP)

The material is also referred to as recycled concrete fines (RCF). Although many studies suggest that the reuse RCA in concrete is feasible, it is in practice often limited to coarse grains [16]. It is presented that the fine fraction of RCA has limited application due to its larger water absorption, which can mangle the fresh and hardened properties of concrete [17]. This is attributed to old mortar attached to the fine grains and high surface area. Quality of old mortar results in a weak interface transition zone (ITZ) and influences the matrix/filler bonding quality in negative way [18]. Some researchers considered the use of recycled concrete fines as a cement replacement, after a thermal treatment. The results show that preheated RCF can take on the rehydration reactivity [19]. Florea et al. conclude that recycled concrete fines, even without thermal treatment, can replace cement without loss of strength, provided that the substitution level is lower than 20% [20]. Braga et al. also show that the reduction of the cement content in mortar can be compensated by the presence of crushed concrete fines in the sand [21]. Reference [16] propose the use of RCF with particles diameter lower than 80 μm as mineral cement substitution; the compressive strength of mortars with RCF was found to be at least equal to that of limestone filler (LF) mortars. Authors state that Portland cement could be substituted by RCF by up to 25% without altering properties of mortars, although activity coefficients of RCF are higher at the early age than those of LF.

Research concerning the use of RCP as a partial replacement of natural sand shows following: reduction in early-age strength, reduction in modulus of elasticity (by 15–20%), and increase in drying shrinkage (by about 40%) [22]. A reduction in compressive strength and increased shrinkage also occurred in the study by [23], who show that using 25% and up to 100% RCF as a replacement of sand results in a reduction in concrete compressive strength of about 15% and 30% respectively, or in the studies by [24,25], who state that increase in drying shrinkage and the reduction in durability seem to be a main impact of utilizing recycled concrete fines in concrete. On the other hand, Evangelista and Brito [26] report that the application of recycled concrete fines does not compromise the mechanical properties of the concrete if substitution up to 30% is maintained. A positive impact is reported by [27] as well, who achieved the maximum compressive strength of the self-compacting concrete (SCC) by using 25–50% RCP as a replacement for river sand, or in the study [28], where improvement of strength development of SCC is presented by supplement of fine recycled concrete up to 40%. The authors suggest the view that fines can act as a nucleation site for new hydration products, as well as they can cause a tightness of the microstructure.

### 1.4. Glass Powder (GP)

Different types of glass at the micro–nano level are tested for building mixtures, including fibers and particles [29,30]. Research works indicate that glass has a chemical composition and phases comparable to traditional SCMs [30]. A process of reducing glass particles, for enhancement of their reactions with cement hydrates, can lead to environmental and economic benefits when cement is partially replaced by milled waste glass for production of concrete [31]. The pozzolanic activity, as is described by [32], is specifically associated with the grain size, although authors give different values to achieve this feature. According to [33], mechanical activation by in high speed grinding mills can support milling process to increase a pozzolanic activity. When blended cement with GP was mechanically activated, an increase in the rate of hydration at an early age was observed, while there was no negative effect on the final properties.

Crushed glass used as aggregates is prone to alkali–silica reaction (ASR) expansion in concrete due to dissolution of amorphous silica in the glass by alkaline attack, forming an ASR gel. In contrast, finely ground glass as an SCM would prevent ASR expansion and ultimately exhibit some mitigating effect on ASR expansion [34]. Sizes below 0.30 mm are believed not to trigger ASR; whereas sizes above 0.60 mm are referred to result in significantly deleterious ASR expansion [35]. However, critical particle size is still being discussed in these terms [36].

It is investigated more as the cement replacement to produce glass-powder-blended cement than as a concrete additive. Particle sizes of 0–100 μm are presented by [37,38] to show a pozzolanic reactivity, improvement in the performances of cement-based products, including resistivity to ASR, with replacement levels of up to 20%. The threshold for noticeable pozzolanic reactivity is generally assumed to be 75 μm [39]. Further decreasing the particle size of GP could significantly increase pozzolanic reactivity, as proven in the study by [40]. They report that GP with particle size ranging between 0 and 25 μm showed higher strength activity index and consumed more portlandite than GP with particle size of 25–38 μm and 63–75 μm.

Investigations on the use of GP as micro-filler—a partial replacement of natural sand—were performed as well. Reference [41] proved that GP with particle size of 100–850 μm and in amount of 25% of cement can be used as a proper and efficient inert powder in high-strength concrete (HSC), which is exposed to firing at high temperature. Similarly [6] presents GP (nanoscale, 600 μm) as useful filler for high-performance concrete (HPC), to replace natural inert powders. Reference [42] tested the glass powder GP of 75 μm as cement addition, while aggregates were reduced accordingly. As given here, GP enhances concrete compressive strength, tensile strength and voids ratio for 33 MPa concrete grade, while for 45 MPa concrete grade, these mechanical and physical properties are improved by up to 25.0% glass powder addition.

At present, specific types of concrete are required in the building industry, which require high workability and higher volume of binder paste. These include such kinds of concrete like pumping concrete, easy-workable concrete, SCC, HPC, and fair-faced concrete. Compared to conventional concrete, those concretes need higher volume of paste so that a larger amount of fine particles is allowed. The amount of fine particles (<250 μm), including cement, mineral powdery additives, and relevant portion of fine aggregate, is one of the technological parameters of concrete. While the content of fine components in classic vibrated concrete must not exceed 550 kgm^−3^ (in case of concretes having D_max_ = 16 mm), for SCC, the maximum limit range of fine particles is 600–650 kgm^−3^ [43]. A high volume of clear cement binder may cause problems in certain types of structures, including an increase in the heat of hydration, the shrinkage potential, and finally also the cost of construction [44]. Therefore, mineral powders can effectively solve a part of relevant problems. Due to their grain size, rheological properties, and lower production of hydration heat, fine additives (both reactive and inert) are commonly used in those concretes to improve and maintain the workability, to regulate the cement content, and to control the properties of hardened concrete.

### 1.5. Research Significance

This study is focused on a highly required type of concrete—concrete with a larger amount of paste to ensure good workability and a smooth surface, along with sufficient physical-mechanical performance. Concretes in which a large portion of fine natural aggregate is replaced with inert mineral powders would offer both economic and ecological benefits for the concrete industry. Moreover, using the powders having potential pozzolanic effect could have positive extra effect on the properties of concrete. However, a large amount of fine particles can also bring an adverse effect on the properties of concrete, which can occur after a longer period of time. There are quite a few publications on the really long-term properties of concrete with the addition of brick, concrete, and glass powders; most publications declaring this usually show the results of concrete at the age of 90 days, 180 days, or maximum 1 year. This paper describes the study on how the high amount of those mineral powders affects the properties of concrete even in a three-year period. It is shown that even after the first year the results of the samples may turn significantly.

## 2. Materials and Methods

### 2.1. Materials

The three following paragraphs describe the properties of materials used.

Portland cement (PC) conforming to EN 197-1 type /B-S 32.5 R (blended cement having blast furnace slag additive and being of 32.5 strength class). Particle size distribution is shown in Figure 1.Natural aggregate (NA) 0/4 and 4/8, density = 2650 kg/m^3^. Fraction 0/4 displayed 8% portion of fine particles (less than 250 μm).Polycarboxylate type of plasticizer + air entraining admixture.Micro-fillers (MF): brick powder (B), concrete powder (C) and glass powder (G) were prepared by crushing the coarse material in laboratory jaw-crusher BCD-32, Brio Hranice, Hranice, Czech Republic, followed by the mechanical separation of fine particles, using 250 μm sieve. Their chemical composition is given in Table 1; particle size distribution is shown in Figure 2, Figure 3 and Figure 4.

As for the chemical character, all micro-fillers have a high content of SiO_2_. Concrete and glass powders are of similar character in terms of SiO_2_ + Al_2_O_3_ + Fe_2_O_3_ content (app. 60%). The brick powder only exceeds the 70% limit for potential pozzolanic properties. Concrete powder has significantly higher amount of CaO than the other two powders. 

As for the grain size in terms of d(0.5), both the brick and glass powders have similar character as the cement, with d(0.5) = 29.385 μm, 26.724 μm, and 26.680 μm respectively. The grain size of concrete powder is higher; d(0.5) = 47.072 μm. The investigated powders have much more portion of coarser grains than the cement, as can be seen by d(0.9) values: 67.866 μm compared to 126.019 μm, 196.640 μm, and 211.693 μm respectively. This is a deliberate consequence of separating particles using 250 μm sieve to obtain grain sizes with a micro-filler character. However, the powders contain a significant portion of particles, giving them a possible pozzolanic character; if the theoretical limit of 100 μm is considered, the powders have the following amounts of these particles: B—85%, C—69% and G—80%.

### 2.2. Mixture Proportioning

Four mixtures of high volume of mortar and high consistency (700 mm by flow test using Abrams cone without compaction—F6 according to EN 206) were designed. The control mixture (C-M) with no powder and three mixtures with different types of powders: brick, concrete, and glass (M-B, M-C and M-G) were considered. As the current research aims to investigate of the effect of a high amount of micro-fillers on the properties of concrete, the cement content, micro-fillers content, and flow were kept constant in all mixes. However, the amount of fine particles creating the volume of matrix/mortar (all particles less than 250 μm, including relevant portion of 0/4 aggregate), is higher in mixtures with powders, as they were applied as 40% replacement of 0/4 aggregate by volume. This is 346 kg, which represents 93% of cement amount by mass. Thus, the amount of fine particles in control mixture C-M reached 456.9 kg (8% of 0/4 = 86.9 kg + CEM 370 kg), while in mixtures having mineral additives M-B, M-C and M-G it reached 768.2 kg (8% of 0/4 = 52.2 kg + CEM 370 kg + MF 346 kg). The amount of water was varied slightly to keep constant consistency; the water demand, as well as the dosage of chemical admixtures was tested previously, and results are given in [45]. For the current experiment, the amount of both plasticizer and air-entraining admixture in C-M was optimized. Actual mix proportions are given in Table 2. The air-entraining admixture was used to improve the durability (freeze-thaw resistance) of mixtures.

Sample handling and testing. Concrete mixtures were mixed in a laboratory mixer. After the fresh concrete flow properties were tested, it was cast into the 150 × 150 × 150 mm cube molds, as well as into the 40 × 40 × 160 mm beam molds. Samples were removed from the molds after one day and further cured under controlled laboratory condition (25 ± 2 °C temperature and 95% ± 5% RH humidity) prior to the tests up to 1 year. Samples were then stored in standard laboratory conditions: temperature (25 ± 2 °C temperature and 60% ± 5% RH humidity) up to the tests at two and three years. Three samples for each property were tested, the results shown being the arithmetic mean.

Following properties reflecting a durability aspect of materials were tested according to EN standard methods: density ρ (kg m^−3^)—EN 12390-7, compressive strength fc (MPa)—EN 12390-3, flexural strength ff (MPa)—EN 12390-5, and coefficient of water absorption due to capillary action (after 90 min of soaking) C (kg m^−2^)—EN 1015-18.

## 3. Results and Discussion

Results of both compressive and flexural strength are given in Figure 5. Results of density and water absorption coefficient due to capillary action are given in Figure 6 to see their relationship directly.

To analyze the impact of different micro-fillers on the properties of concretes over a three-year period, two aspects are presented in tables: annual percentage increase (+)/decrease (−) in tested properties including total change after three years of curing (Table 3) andpercentage change in the properties of micro-filler-based samples compared to the control one (Table 4). In the experiment, the following values of properties were found:

Compressive strength: The values after 28 days of setting and hardening range from 29 to 48 MPa, while values after three years range from 54 to 93 MPa. Based on the 28-day results, samples can be classified into strength classes C 20/25 (C-M), C 25/30 (M-G) and C 35/45 (M-B and M-C) in accordance to standard EN 206.

Flexural strength: The values after 28 days of setting and hardening range from 5.0 to 8.4 MPa, while values after three years range from 6.7 to 11.1 MPa. The best values were achieved by M-C, with M-B results being very close.

Density: the values after 28 days of setting and hardening range from 2180 to 2370 kg m^−3^, while values after three years range from 2150 to 2300 kg m^−3^. The values in all ages fall in the range 2000–2600 kg m^−3^, meaning that the samples can be classified as a normal weight concrete in accordance with EN 206.

Water absorption coefficient due to capillary action: the values after 28 days of setting and hardening range from 1.79 to 2.24 kg m^−2^ while values after three years range from 1.62 to 2.43 kg m^−2^.

### 3.1. Changes of Properties in Time

#### 3.1.1. Generally

Looking at the figures, it is possible to see a continuous change of properties over time, and the correlation between properties is good. If we omit the results within 28 days with the biggest changes in properties and focus on the behavior of the samples at a later time, we can state the following: all samples show increasing values of both compressive and flexural strength with time during the whole tested period, while in principle, the greatest increase was recorded after the first year of curing (except for the compressive strength of C-M, which has the greatest increase in the third year, and flexural strength of M-B, with the highest increase also in the third year). Increasing tendency of strength with time is fully in accordance with the well-known knowledge of hydration process of cement-based mixtures, as presented, for example, in [46]. This effect is generally attributed to the gradual formation of hydration products over time, which causes gradual strengthening. Capillary absorption, which decreases due to the likely denser structure, and volume density, which is increasing for the same reason, also correlate well with the aforementioned strength of the first year of treatment. In the second and third years, the tendency of changes in bulk density and capillary absorption changes, while they are in good correlation with each other. The bulk density decreases, but the changes are only modest, with the capillary absorption increasing at this time (with the exception of M-G, which continues to decline slightly).

#### 3.1.2. Compressive Strength

Samples with micro-fillers achieved bigger overall increase in compressive strength from 28 days to three years than the control mixture (the biggest increasing tendency showed M-G: 114.5%), while a sharper increase was detected in the first-year period than during the next two years. The control mixture C-M showed the sharpest increase in the third year, while overall increase in compressive strength in the there-year period was the smallest for all samples: 86%. The stronger development of strength in the third year may be attributed to the higher presence of GGBFS in the cement (CEM II/B-S), the activity of which is only shown later. The increasing tendency of compressive strength with time, when the effect of additives was tested, was also recorded by [28] (recycled concrete fines as a 40% of cement mass addition), as well as by [42] (glass powder as cement addition up to 25% for 45 MPa concrete grade), or by [13] for brick powder. However, they tested samples only up to 180, 56, and 90 days respectively. As for the sharper increase in the strength of M-G in the first year of curing, similar results are reported by [47]; the study also shows a stronger increase in the compressive strength of mortars having glass powder addition between 28 and 200 days of testing than between 200 and 400 days. Authors attribute this to greater pore network refinement of mortars with waste glass powder in comparison to CEM I ones in the medium and long term. This is probably because of the consecutive formation of solid phase as a result of pozzolanic reaction of glass powder.

#### 3.1.3. Flexural Strength

The percentual increase of flexural strength with time is not as high as that of compressive strength; the overall increase from 28 days to three years runs between 12.7% (M-G) and 34% (C-M). Unlike the case of compressive strength development, the control sample showed sharper increase in the first-year period (14%) than during next two years, as well as it achieved the highest overall increase up to three years (34%). Also the behavior of the M-G sample, which achieved the biggest increase in compressive strength in the first-year period as well as overall after three years, is different; the sample achieved the lowest increase in flexural strength of all samples in both time periods: 6.3% and 12.7% respectively. 

#### 3.1.4. Density

All samples show the tendency of increase in density up to 365 days—this is in line with well-known knowledge on densifying of concrete with time. However, after the period of one year, the values start to fall slightly. This applies to all samples, including the control one. However, even the greatest change in density, which was detected at 4.7% (M-B), is not of great significance; thus, density can be characterized as relatively stable. M-G was found to have the lowest overall decrease in density, at 1.7%.

#### 3.1.5. Water Absorption Coefficient due to Capillary Action

All samples show the tendency of decreasing absorption coefficient up to 365 days. Overall, after three years of curing, M-G was found to have the biggest decrease in absorption coefficient: −26.0%, while M-B showed significant increase: 27.2%.

As for lack of studies on the long-term water absorption coefficient due to capillary action, the study on porosity can be taken for reference purposes [47]. The study shows that porosity was reduced from 28 to 200 days for all the studied mortars (having 10 and 20% of clinker replacement by glass powder) and remained almost constant or slightly increased up to 400 days. This behavior is similar to what is observed in our case—reduction of water absorption (being directly linked to porosity) up to 365 days after which there is no change or growth. 

However, since we have been studying samples for a longer time, we have found that after the period of one year, the samples showed different behavior: while C-M and M-G were very similar, showing practically constant values, values of both M-C and M-B began to grow, showing a quite significant increase. These results first correspond to results of density as for the rate of its increase after the first year (the biggest one was found for M-C and M-B, too), and secondly they are probably due to the character of both concrete and brick powder as for water absorption ability, which takes effect after a long time.

Our results up to one year are in agreement with [48], who suggest replacing 5.5% of the fine aggregate fraction by various mineral additives including fly ash, sandstone powder, limestone powder, and silica fume. They observe a positive effect on sorptivity along with compressive strength for 28-day old concrete, since micro-filler materials having fine particles fill both the interfaces and the bulk paste. However, we have found a change in sample behavior over a long-term period for brick and concrete powders, as described above.

### 3.2. Changes Compared to the Control Mixture C-M

#### 3.2.1. Generally

All samples containing micro-fillers achieved better results (higher compressive and flexural strength, higher density, and lower water absorption coefficient) than the control one C-M. The only exception is absorption coefficient of M-C and M-B after the first year. The results are evident even though C-M had slightly lower w/c ratio, so considering only the amount of water aspect, it would be expected to have the highest strength. Therefore, the positive effect of micro-fillers, as used in the experiment presented, seems to be convincing; their beneficial effect outweighs the adverse effect of the higher amount of water. 

#### 3.2.2. Compressive Strength

The differences between the values of micro-filler-based samples and the control sample are significant, mainly after a longer time. Compared to the C-M, the differences are more significant for both M-B and M-C samples: 66.6% and 58.6% respectively in 28 days, and 72.2% and 67.6% respectively in three-year period of curing.

Unlike most publications that describe lower strength of mixtures with pozzolanic additives at an earlier age, our research showed those mixtures as having higher strength even in 28-days. This may probably be caused by the arrangement of mixtures—powders as additives to cement and replacement to sand, as well as by their amount, causing the change in aggregate: binder proportion (see Table 2). Ortega et al. [47] describe the delay in pozzolanic reactions when the clinker was replaced by glass powder, compared to only clinker hydration in relation to the compressive strength parameter, while the development of those pozzolanic reactions allows a progressive rise of this strength later, at 200 and 400 days. On the other hand, a similar positive effect as ours was found by [8], who also applied the BP as a sand replacement (10–100 %). Here, all samples also achieved higher 28-day values of compressive strength than that of control sample, while 40% replacement level was found to be the best. This is the same as our sand replacement level. Such results are addressed to both the filler and pozzolanic effects. As the BP and cement have different particle-size distributions, the BP particles can improve a tightness of the microstructure, while both the amorphous phases in BP and active SiO_2_ are activated by the cement hydration product Ca(OH)_2_(CH), resulting in following chemical reaction. Under higher dosage of BP, the availability of amorphous phases and active SiO_2_ is improved, resulting in a greater rate of the pozzolanic reaction and new hydration phases formation. The same can probably be attributed to our results; as we had the brick powder of d(0.9) = 126.019 μm, so some portions of grains are coarser than the value for pozzolanic reaction, as well as SiO_2_ + Al_2_O_3_ + Fe_2_O_3_ content, which is more than 70%.

Positive results of sample M-C may be attributed to higher content of RCP together with the grain-size d(0.9) = 196.64 μm. This falls into the range given by [17], who investigated the potential use of RCP as microsilica sand substitute in the production of engineered cementitious composites. Compared to the control mixture without RCP, mixture with fine RCP of 0–300 μm maintained the same level of compressive strength. Authors attribute this to self-cementing properties of RCP, as has been suggested in literatures [10,27]. Although the inclusion of fine RCP introduces more interface transition zones (ITZ), it may also have a higher content of un-hydrated cement, which improves the ITZ of the resulting mixture. They achieved a maximum compressive strength by using 25–50% RCP. Similarly, [28] presents SCC with the addition of fine recycled concrete of up to 40% of cement mass, while reporting an improvement of the strength compared to the control sample in all tested periods (3, 7, 28, 60, 90 and 180 days); they attribute it to RCP, acting as a nucleation site for new hydration products and densifying the microstructure.

Sample with glass powder M-G achieved the lowest values of compressive strength of all micro-filler samples, although still much better than that of control mixture C-M. Aliabdo et al. [42] also present increase in compressive strength compared to the control sample for 45 MPa concrete grade. However, they tested samples having glass powder of 75 μm as an addition of only up to 25% of cement mass. As presented in [32], the pozzolanic activity of GP comes as its positive feature; the amorphous silica (SiO_2_) in the GP reacts with the portlandite [Ca(OH)_2_] generated during cement hydration, and forms gels of calcium silicate hydrate (C-S-H). The process is usually associated with the fineness of the GP.

It is likely that slightly worse results of M-G, compared to both M-B and M-C, are because of coarser character: while d(0.9) of brick powder and concrete powder was 126.019 μm and 196.640 μm respectively, d(0.9) of glass powder was 211.693 μm—i.e., some portion of grains are bigger than 100 μm (18%), the limiting value generally attributed to pozzolanic properties of glass particles. It can be stated here that about 82% of G was probably involved in the pozzolanic reaction, while the rest (18%) worked as an inert micro-filler. According to [6], GP of nanoscale to 600μm can act as filler in HPC or UHPC when using as substitution for natural fillers, to achieve better efficiency.

#### 3.2.3. Flexural Strength

As in the case of compressive strength, all samples containing micro-fillers achieved higher flexural strength than that of control sample during the whole tested period; the same positive effect of micro-fillers can be observed here. This can be attributed to the way the micro-fillers act in the cement matrix, as described in [49]: if strong and stiff particles are introduced into the cement matrix (characterized by ion-covalent character of CSH gel, leading to complex crystal and amorphous structure, associated by brittleness), they can act as ‘crack-stoppers’. 

The order of samples correlate well to order in compressive strength; the M-B and M-C samples are closer to each other while having the highest values, followed by M-G. M-B and M-C have the biggest difference compared to the C-M (60.0% and 68.0% respectively in the 28-day, and 44.8% and 65.7% respectively in the three-year period of curing). The difference between the M-G and C-M is 26.0% after 28 days, but it is only 6.0% in the long term. Similar to the compressive strength, Li et al. [8] found BP to also have positive effect on the flexural strength, while being the replacement of sand from 10–100%. All samples achieved higher 28-day values of flexural strength than that of control sample, while 40% replacement level was found to be the best. 

#### 3.2.4. Density

All the samples containing micro-fillers achieved higher density compared to the control one, while mixture with concrete powder M-C achieved the highest values in whole the tested period. All the percentage differences between C-M and micro-filler-based mixtures run between 2.8–8.7%.

This correlates with the knowledge on the effect of additives for increase in density [1,8].

#### 3.2.5. Water Absorption Coefficient due to Capillary Action

All the samples containing micro-fillers achieved lower (better) coefficient compared to the control one, up to a one-year period, while mixture with concrete powder M-C achieved the best values. In the second year, the absorption of M-C was found to be increasing, becoming practically the same as that of both C-M and M-G, while the M-B was even higher. This trend continued afterwards, while after three years, the difference between both C-M and M-C was +9.4%, and between C-M and M-B it was even +42.9%. Out of all samples, only M-G showed lower absorption than the control sample during the whole testing period, i.e., in the long-term point of view. This can be attributed to the absorptive character of both the brick and the concrete powder. This absorptive character was proven, for example, by [50], who observes for RCP-based concrete increasing water absorption with increasing amount of powder.

## 4. Conclusions

This paper analyzed the impact of a high dosage of three kinds of micro-fillers on the properties of concrete over a three-year period. Microfillers were applied as 40% replacement of 0/4 aggregate by volume. The control mixture with no powder and with aggregate:binder ratio = 0.65:0.35 was compared with three mixtures having brick, concrete and glass powders (BP, CP and GP), and with aggregate:binder ratio of 0.48:0.52. The following options can be drawn from the results obtained in this investigation:The importance of long-term testing was proven; significant changes in values were observed in each of the three years of testing.Samples having high dosage of micro-fillers and thus a higher binder volume:Achieved good capillary water absorption performance up to one year of curing; however, long-term testing has shown a change in the behavior of samples with BP and CP as well. These samples seem to be more sensitive to changes in property values after a longer curing period.Achieved significant increase in strength characteristics over the three-year period, while the greatest increase was recorded after the first year of curing.Achieved excellent values of compressive strength: from 31 to 48 MPa in 28 days, and from 67 to 93 MPa in three years.Achieved excellent values of flexural strength: from 6.3 to 8.4 MPa in 28 days, and from 7.1 to 11.1 MPa in three years.Contrary to application as a cement substitution, micro-fillers applied as a cement addition gave rise to higher strength characteristics than the control mixture even in short time of 28-days.Both samples with BP a CP achieved the biggest strength values; however, due to better performance in durability parameters (capillary water absorption coefficient and density), sample prepared with GP can be identified as having the biggest potential for intended use.

## Figures and Tables

**Figure 1 materials-12-03421-f001:**
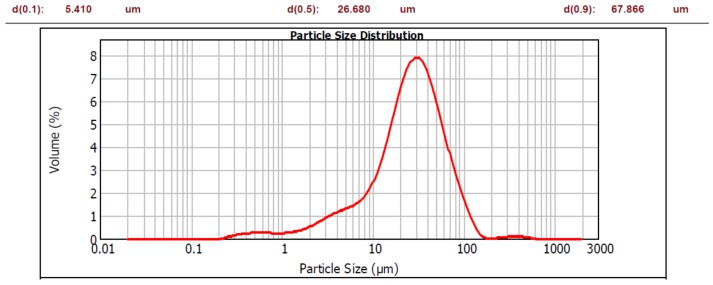
Particle size distribution of cement.

**Figure 2 materials-12-03421-f002:**
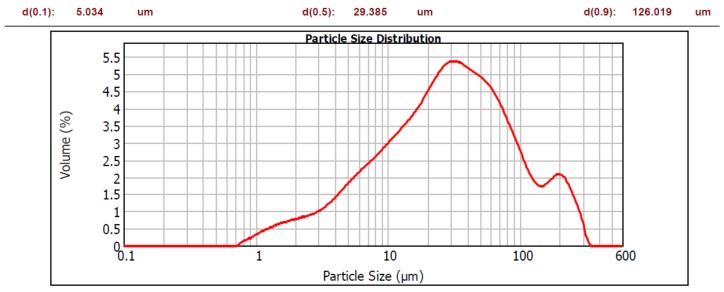
Particle size distribution of brick powder.

**Figure 3 materials-12-03421-f003:**
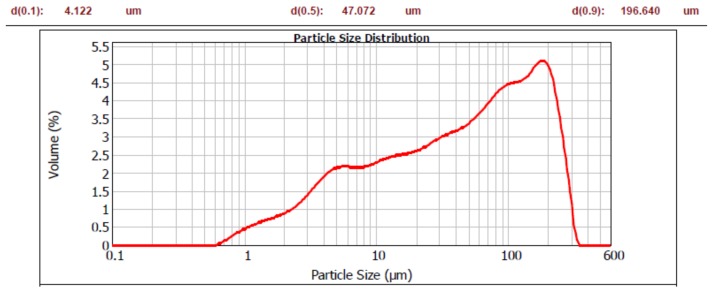
Particle size distribution of concrete.

**Figure 4 materials-12-03421-f004:**
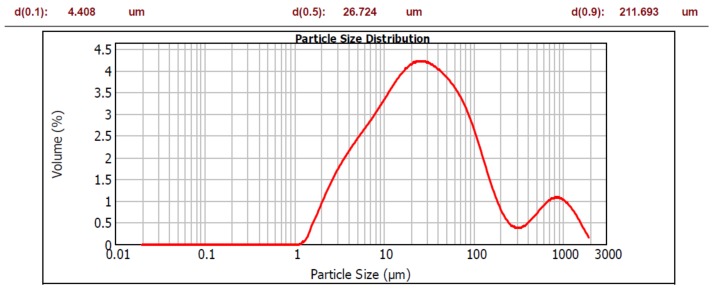
Particle size distribution of glass powder.

**Figure 5 materials-12-03421-f005:**
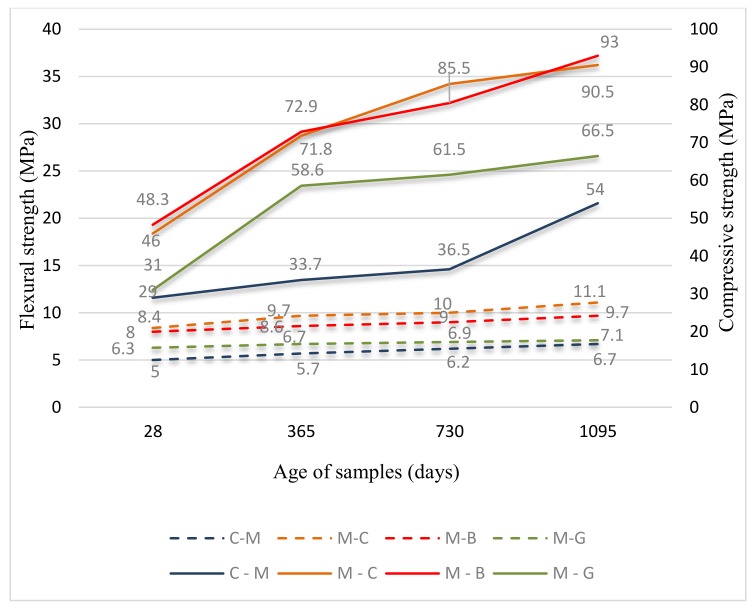
Results of compressive and flexural strength.

**Figure 6 materials-12-03421-f006:**
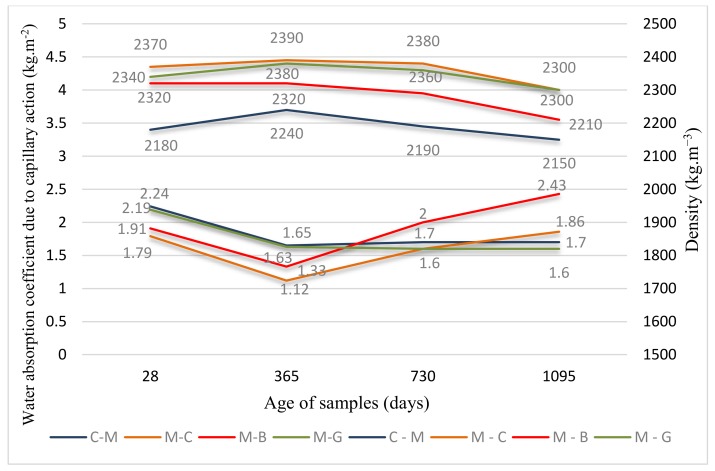
Results of density and water absorption coefficient due to capillary action.

**Table 1 materials-12-03421-t001:** Chemical composition of tested powders.

Materials	CaO (%)	Al_2_O_3_ (%)	SiO_2_ (%)	Fe_2_O_3_ (%)	MgO (%)	SiO_2_ + Al_2_O_3_ + Fe_2_O_3_ (%)
Brick powder	1.6	15.4	73.0	4.3	2.1	92.7
Concrete powder	28.4	6.4	53.9	2.8	4.5	63.1
Glass powder	5.2	0.9	59.5	0.1	4.2	60.5

**Table 2 materials-12-03421-t002:** Mixture proportions.

Components	Units	Mixtures			
		Control Mixture C-M	Brick Mixture M-B	Concrete Mixture M-C	Glass Mixture M-G
Cement II/B-S 32.5 R	kg	370	370	370	370
Natural aggregate 4/8	kg	694	694	694	694
Natural aggregate 0/4	kg	1086	652	652	652
Micro-filler	kg	-	346	346	346
Water	kg	195	218	218	215
Water/cement ratio w	-	0.53	0.59	0.59	0.58
Plasticizer	%	1.5	1.5	1.5	1.2
Air entrainment admixture	%	0.4	0.4	0.4	0.4
Volume of aggregate: paste	m^3^	0.65:0.35	0.48:0.52	0.48:0.52	0.48:0.52

**Table 3 materials-12-03421-t003:** Annual percentage increase (+)/decrease (−) of property values.

Parameter	Sample	1st Year * (%)	2nd Year (%)	3rd Year (%)	Total Change after 3 Years (%)
Density	C-M	+0.8	−0.4	−3.4	−3.0
	M-B	0	−1.3	−3.4	−4.7
	M-C	+0.8	−0.4	−3.4	−3.0
	M-G	+1.7	−0.9	−2.5	−1.7
Compressive strength	C-M	+16.2	+9.7	+60.3	+86.2
	M-B	+50.9	+15.7	+25.9	+92.5
	M-C	+56.1	+29.8	+10.8	+96.7
	M-G	+89.0	+9.4	+16.1	+114.5
Flexural strength	C-M	+14.0	+10.0	+10.0	+34.0
	M-B	+7.5	+5.0	+8.8	+21.3
	M-C	+15.4	+3.6	+13.1	+32.1
	M-G	+6.3	+3.2	+3.2	+12.7
Water absorption coefficient due to capillary action	C-M	−26.3	+2.2	0	−24.1
	M-B	−30.3	+35.0	+22.5	+27.2
	M-C	−37.3	+26.7	+14.5	+3.9
	M-G	−25.5	−1.4	0	−26.9

* The values were calculated as the change pertaining to the particular year, comparing 28-day values.

**Table 4 materials-12-03421-t004:** Percentage increase (+)/decrease (−) of property values of micro-fillers-based samples comparing to the control mixture C-M, per year.

Parameter	Sample	28-Days (%)	1st Year (%)	2nd Year (%)	3rd Year (%)
Density	M-B	+6.4	+3.6	+4.6	+2.8
	M-C	+8.7	+6.7	+8.7	+7.0
	M-G	+7.3	+6.3	+7.8	+7.0
Compressive strength	M-B	+66.6	+116.3	+120.5	+72.2
	M-C	+58.6	+113.1	+134.2	+67.6
	M-G	+6.9	+73.9	+68.5	+23.1
Flexural strength	M-B	+60.0	+50.9	+45.2	+44.8
	M-C	+68.0	+70.2	+61.3	+65.7
	M-G	+26.0	+17.5	+11.3	+6.0
Water absorption coefficient due to capillary action	M-B	−4.3	−19.4	+17.6	+42.9
	M-C	−20.1	−32.2	−5.9	+9.4
	M-G	−2.2	−1.2	−5.9	−5.9

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
