# Peer review of "The Effect of a High Amount of Micro-Fillers on the Long-Term Properties of Concrete"

_materials, 2019, doi:10.3390/ma12203421_

Round 1

Reviewer 1 Report

This paper presents various experimental data as a manuscript that analyzes the effect of micro-fillers such as brick, concrete and glass powder on the long-term properties of concrete.

However, authors should provide additional explanation for the following:

(Table 2) The unit water and w/c (water-cement ratio) of the mixture using the micro-filler is about 10% higher than that of the control mixture (C-M), which may affect the strength. Rather, the w/c should be the same and the plasticizer should be adjusted to control the fluidity.

(Fig.5) The compressive strength of C-M mixture has increased rapidly after 730 days, which is different from the general compressive strength evolution trend. The authors should analyze the cause for this.

Author Response

Reviewer 1:

Firstly, we deeply thank the reviewer for his valuable and constructive comments which do help to improve the quality of our paper. The previous manuscript has been revised according to the comments and the detailed responses are given below.

For changes in the manuscript, please see the attached manuscript and follow the text in green.

This paper presents various experimental data as a manuscript that analyzes the effect of micro-fillers such as brick, concrete and glass powder on the long-term properties of concrete.

However, authors should provide additional explanation for the following:

1          (Table 2) The unit water and w/c (water-cement ratio) of the mixture using the micro-filler is about 10% higher than that of the control mixture (C-M), which may affect the strength. Rather, the w/c should be the same and the plasticizer should be adjusted to control the fluidity.

We've chosen one of two possible ways how to manage experiment: mixtures with the same w, or mixtures with the same consistency. The principle of testing the mixtures under the same (almost) consistency, is quite often used, some examples are given below.  It reflects the practical aspect, when defined consistency is to be achieved.

Omar S. Baghabra Al-Amoudi et al: Performance of blended cement concretes prepared with constant workability: (“The water content in the concrete mixtures was adjusted to obtain workability in the range of 75–100 mm slump…”)

https://www.sciencedirect.com/science/article/pii/S0958946510001630

Haoxin Li et al: Study on utilization of red brick waste powder in the production

of cement-based red decorative plaster for walls (“Different amounts of mixing water were added to maintain a similar level of workability”).

https://www.sciencedirect.com/science/article/pii/S095965261630628X

Ahmed O. Mashaly et al: Performance of mortar and concrete incorporating granite sludge as cement replacement: (“The cement composites pastes were prepared by using above mentioned proportions of GS substituting for OPC and mixed with the optimum water obtained by water consistency …”)

https://www.sciencedirect.com/science/article/pii/S0950061818305294

Moreover, the amount of water is lower only for the control mixture (C-M); the other mixtures with additives have practically the same w. Thus, the results could be unclear if C-M achieved better parameters than other samples. In our case, despite the lower water content, C-M results are worse. Therefore, it can be assumed that even with the same w, this would be the case, even to a greater extent.

This aspect is mentioned in the discussion of results, chapter 3.2.1

2          (Fig.5) The compressive strength of C-M mixture has increased rapidly after 730 days, which is different from the general compressive strength evolution trend. The authors should analyze the cause for this.

The probable cause has been added into 3.1.2

Reviewer 2 Report

In my opinion, this is a very interesting work, focused in the use of micro filler on the properties of concrete. I think this article is novelty and the authors include in the document the explanations of this adequate study, in which applying different mixtures of brick powder, concrete powder and glass powder.

It could be improved including explanations of the possible reaction silica-alcali in the batch made with glass powder.

Also, they can add shrinkage measures because it is a very important property on the long term. Have you made these measures?

In each mix, how many specimens were manufactured and tested by property?

Please, change in the text granularity by size distribution

Author Response

Reviewer 2:

Firstly, we deeply thank the reviewer for his valuable and constructive comments which do help to improve the quality of our paper. The previous manuscript has been revised according to the comments and the detailed responses are given below. For changes in the manuscript, please see the manuscript attached and follow the text in blue.

1          In my opinion, this is a very interesting work, focused in the use of micro filler on the properties of concrete. I think this article is novelty and the authors include in the document the explanations of this adequate study, in which applying different mixtures of brick powder, concrete powder and glass powder.

It could be improved including explanations of the possible reaction silica-alcali in the batch made with glass powder.

This aspect has been added to Introduction as well as relevant references.

2          Also, they can add shrinkage measures because it is a very important property on the long term. Have you made these measures?

Unfortunately not. Only visual observations of macroscopic cracks were performed, with negative findings (without cracks). We are planning to extend our testing in this respect in the future. We hope that the article still contains useful information for potential readers.

3          In each mix, how many specimens were manufactured and tested by property?

Three samples for each property were tested, the results shown being the arithmetic mean.

The information is added into the article – chapter 2.

4          Please, change in the text granularity by size distribution

Corrected.

Reviewer 3 Report

The paper deals with the use of micro-fillers based on crushed brick, crushed concrete or GRP for concrete production. Authors present interesting results about the long-term tests, highlighting that the properties of the mixtures cannot be evaluated only at 28 days, especially when micro-fillers are used. On the other hand, I have several puzzlement regarding the experimental design. In particular, authors should explain why they decided to attain the same workability class at the end of the mixing by varying both the water to cement ratio and the superplasticizer dosage. In this way, it is impossible to correlate the variation of the physical and elasto-mechanical properties of concrete with the use of micro-fillers instead of natural sand. Moreover, authors should explain why they decided to use an air entraining agent and why they use it at different dosages. Finally, several supplementary tests could be performed in order to better understand both the long-term behaviour and the microstructure of micro-filler based concrete (XRD, TGA, SEM and so on).

Other suggestions can be found on the attached PDF file.

For the abovementioned reasons, I recommend to accept this paper after major revision.

Author Response

Firstly, we deeply thank the reviewer for his valuable and constructive comments which do help to improve the quality of our paper. The previous manuscript has been revised according to the comments and the detailed responses are given below. For changes in the manuscript, please see the manuscript attached and follow the text in gray.

The paper deals with the use of micro-fillers based on crushed brick, crushed concrete or GRP for concrete production. Authors present interesting results about the long-term tests, highlighting that the properties of the mixtures cannot be evaluated only at 28 days, especially when micro-fillers are used. On the other hand, I have several puzzlement regarding the experimental design.

1          In particular, authors should explain why they decided to attain the same workability class at the end of the mixing by varying both the water to cement ratio and the superplasticizer dosage. In this way, it is impossible to correlate the variation of the physical and elasto-mechanical properties of concrete with the use of micro-fillers instead of natural sand.

We've chosen one of two possible ways how to manage experiment: mixtures with the same w, or mixtures with the same consistency. The principle of testing the mixtures under the same (almost) consistency, is quite often used, some examples are given below.  It reflects the practical aspect, when defined consistency is to be achieved.

Omar S. Baghabra Al-Amoudi et al: Performance of blended cement concretes prepared with constant workability: (“The water content in the concrete mixtures was adjusted to obtain workability in the range of 75–100 mm slump…”)

https://www.sciencedirect.com/science/article/pii/S0958946510001630

Haoxin Li et al: Study on utilization of red brick waste powder in the production

of cement-based red decorative plaster for walls (“Different amounts of mixing water were added to maintain a similar level of workability”).

https://www.sciencedirect.com/science/article/pii/S095965261630628X

Ahmed O. Mashaly et al: Performance of mortar and concrete incorporating granite sludge as cement replacement: (“The cement composites pastes were prepared by using above mentioned proportions of GS substituting for OPC and mixed with the optimum water obtained by water consistency …”)

https://www.sciencedirect.com/science/article/pii/S0950061818305294

Moreover, the amount of water is lower only for the control mixture (C-M); the other mixtures with additives have practically the same w. Thus, the results could be unclear if C-M achieved better parameters than other samples. In our case, despite the lower water content, C-M results are worse. Therefore, it can be assumed that even with the same w, this would be the case, even to a greater extent.

This aspect is mentioned in the discussion of results, chapter 3.2.1

 2          Moreover, authors should explain why they decided to use an air entraining agent and why they use it at different dosages.

The explanation as for reason is added to the article (freeze-thaw resistance; please note the results of testing this parameter are not included in this article).

Question of dosage: as given in the article, some optimisation in composition of mixtures has been made comparing our previous experimental program, but now we can see that only plasticizer is mentioned in the article. It is only because we have forgotten to pay attention to the change in air entraining admixture dosage which has occurred practically, and we have not mentioned it neither in the table nor in the text.

Data are corrected now – see “Mixture proportioning” chapter and tab. 2.

3          Finally, several supplementary tests could be performed in order to better understand both the long-term behaviour and the microstructure of micro-filler based concrete (XRD, TGA, SEM and so on).

Unfortunately, we do not have such a test available. However, we hope that the article still contains a nice amount of useful information for potential readers.

4          Other suggestions can be found on the attached PDF file.

All of them are accepted – see coloured sections of text/references.

Round 2

Reviewer 3 Report

All the suggestions have been addressed